# Saturation kinetics and specificity of transporters for L-arginine and asymmetric dimethylarginine (ADMA) at the blood-brain and blood-CSF barriers

Mehmet Fidanboylu[1,2], Sarah Ann Thomas [1,2]*

1 Pharmaceutical Sciences Research Division, King's College London, London, United Kingdom,
2 Institute of Pharmaceutical Science, King's College London, London, United Kingdom

* sarah.thomas@kcl.ac.uk

## Abstract

Nitric oxide synthases (NOS) synthesize nitric oxide (NO) from L-arginine in endothelial and neuronal cells. Asymmetric dimethylarginine (ADMA) is a homologue of arginine and an endogenous inhibitor of NOS. As NO is a critical signalling molecule and influences physiological pathways in health and disease, the transfer of arginine and ADMA across the blood-CNS barriers is of interest. Our research group have previously demonstrated the presence of saturable transporters for [$^3$H]-arginine and [$^3$H]-ADMA at the blood-brain and blood-CSF barriers using *in vitro* and *in situ* methods. In this study, we determine the identity and kinetic characteristics of these transporters by means of the *in situ* brain/choroid plexus perfusion technique in anaesthetised mice. Results indicated that [$^3$H]-arginine and [$^3$H]-ADMA could be transported across blood-brain and blood-CSF barriers by the cationic amino acid transporter, system-y$^+$. In contrast to the results obtained with arginine where transport was predominately by a single transport system (system-y$^+$), ADMA delivery to the CNS was more complex and involved multiple transport systems (system y$^+$, B$^{0,+}$, y$^+$L and b$^{0,+}$) suggesting its concentration is tightly regulated. System y$^+$ and system y$^+$L transporters could be involved in the CNS to blood efflux of ADMA that we have previously observed. The half-saturation constant ($K_m$) and maximal influx rate of the saturable component ($V_{max}$) for [$^3$H]-ADMA transport into the frontal cortex was $29.07 \pm 7.19$ μM and $0.307 \pm 0.017$ nmol.min$^{-1}$.g$^{-1}$, respectively, and into the CSF was $30.59 \pm 25.41$ μM and $2.07 \pm 0.38$ nmol.min$^{-1}$.g$^{-1}$, respectively. This information could help explain the arginine paradox providing evidence that ADMA interacts with transporters that can remove ADMA from cells. These removal mechanisms could be stimulated by excess arginine in the plasma resulting in increased NO production. It remains to be seen if arginine supplementation could be used to increase NO production and improve hypoperfusion observed in disease states such as Alzheimer's and stroke.

**Data availability statement:** All relevant data are within the manuscript and its Supporting Information files.

**Funding:** This work was supported by a Biotechnology and Biological Sciences Research Council (BBSRC) centre for integrative biomedicine PhD studentship for Mr Fidanboylu to Dr Sarah Ann Thomas [BB/E527098/1]. https://www.ukri.org/councils/bbsrc/. This research was funded in whole, or in part, by the Wellcome Trust [080268]. https://wellcome.org/. The recipient of this grant was Dr Sarah Ann Thomas. For the purpose of Open Access, the author has applied a CC BY public copyright licence to any Author Accepted Manuscript version arising from this submission. The funders had no role in study design, data collection and analysis, decision to publish, or preparation of the manuscript.

**Competing interests:** The authors have declared that no competing interests exist.

**Abbreviations:** AAPB, L-arginine S-2-(1-acetoxy-n-pentyl)-benzoate; ADMA, asymmetric dimethylarginine or $N^G,N^G$-dimethyl-$L$-arginine; BCH, 2-amino-endo-bicyclo[2.2.1]heptane-2-carboxylic acid; BBB, blood-brain barrier; CAAs, cationic amino acids; CAT, cationic amino acid transporter; CNS, central nervous system; CSF, cerebrospinal fluid; CVO, circumventricular organ; $K_d$, constant of non-saturable diffusion; eNOS, endothelial nitric oxide synthase; $K_m$, half-saturation constant; $V_{max}$, maximal influx rate of the saturable component; $C_{cap}$, mean capillary concentration of the amino acid; NO, nitric oxide; NOS, nitric oxide synthase; $K_{in}$, unidirectional transfer constant; $J_{in}$, unidirectional flux; SD, standard deviation; SEM, standard error of the mean; nNOS, neuronal nitric oxide synthase

## Introduction

There are two barriers between the blood and the central nervous system (CNS). One is called the blood-brain barrier (BBB) and is located at the brain capillary endothelial cell wall. The other is the blood-cerebrospinal fluid (CSF) barrier and is formed by the choroid plexuses and arachnoid membrane. Both barriers serve to protect the sensitive and delicate brain cells from harmful substances in the bloodstream, but at the same time allow oxygen and nutrients into the brain. The selectivity of the barriers is due to the presence of tight junctions, which physically restrict passage between the cells, and the presence of transporters (also called carriers), ion channels and transcytosis machinery, that move substances into and out of the CNS [1]. The barriers also control molecule passage into the CNS by serving as a metabolic (enzymatic) interface.

Over the last 30 years there have been advances in our knowledge of cationic amino acid (CAA) transport. It is now clear that there are five transport systems for CAAs that show differences in site of expression, substrate specificity and dependence upon $Na^+$-gradients. These include the $y^+$, $y^+L$, $b^{o,+}$, $B^{o,+}$ and $b^+$ transport systems [2–4]. CAAs are an absolute requirement for normal brain function and yet little is known about how they enter from blood to brain in the most used laboratory model, the mouse. Of particular interest is arginine, a semi-essential CAA and the exclusive substrate for nitric oxide (NO) synthesis by nitric oxide synthases (NOS), and asymmetric dimethylarginine (ADMA) – an endogenously occurring analogue of $L$-arginine, which inhibits NO production. It is believed that the inhibition of NOS by ADMA stems from the inability of NOS to utilise ADMA as a substrate [5]. There are three isoforms of NOS including endothelial (eNOS), neuronal (nNOS) and inducible (iNOS). Transporters are needed to provide these cells with the arginine required for NO synthesis. NO has been implicated in regulatory and mediatory roles in a variety of different physiological processes and pathological conditions involving the cardiovascular, nervous and immune systems. In particular, eNOS is critical for cardiovascular health as it plays a key role in controlling vascular tone (blood flow), inhibiting inflammation and preventing thrombosis [6]. Transporters for CAA therefore have an intermediate role in regulating vascular tone and blood flow. Importantly, exogenous arginine has been used as a pre-training NO booster to increase blood flow in muscles during training, improve athletic performance and speed up recovery [7,8]. In addition, L-arginine supplementation can reverse cerebrovascular arteriolar vasoconstriction, increase cerebral blood flow and partially reverse cerebral ischemia in experimental cerebral malaria [9,10].

The role of NO in the brain is both complex and far reaching, and any disruption to the delicately balanced processes that control its production can be expected to have severe consequences in the brain. In fact, abnormal levels of CAA (such as ADMA) and their associated molecules (such as NO) are associated with several diseases including Alzheimer's and stroke. Importantly, both Alzheimer's and ischemic stroke are linked to hypoperfusion of brain tissues [11]. Interestingly, a study which used a drug-induced rat model of Alzheimer's disease demonstrated that administering arginine was protective against Alzheimer's by increasing hippocampal NO levels

likely via increased nNOS activity [12]. Arginine supplementation has been shown to ameliorate the choroid plexus pathological changes in an Alzheimer's disease aged rat model [13]. It remains to be seen if these beneficial effects of arginine administration in Alzheimer's models are due, in part, to changes in tissue perfusion. Importantly, a novel arginine-based compound (namely L--arginine S-2-(1-acetoxy-n-pentyl)-benzoate) (AAPB) is in development which has the dual effects of NO-dependent cerebral blood flow improvement and neuroprotection to treat ischemic stroke [14,15].

In this study we build on our previous research studies which identified saturable transporters for [3H]-arginine and [3H]-ADMA at both the blood-brain and blood-CSF barriers [16,17]. Our aim was to explore and compare the identity of the transporters for [3H]-arginine and [3H]-ADMA at the BBB by means of the *in situ* brain perfusion technique in anaesthetised mice and the use of specific transporter inhibitors. In addition, this method allows examination of uptake directly into the CSF and so the characteristics of passage across the blood-CSF barrier, at the level of the choroid plexus, was also investigated. We also determined the kinetic characteristics of [3H]-ADMA flux into the CNS by calculation of the half-saturation constant ($K_m$), maximal influx rate of the saturable component ($V_{max}$) and constant of non-saturable diffusion ($K_d$). Although our earlier study detected some overlap in transporter specificity by arginine and ADMA, we now test the hypothesis that as these CAAs have opposite functions, they may also use different transporters despite their structural similarities. This knowledge is essential, not only for understanding the local and neuronal physiological processes that are mediated by the CAAs these carriers transport, but also due to the widespread influence of NO in physiological pathways in health and disease. In addition, identification of the transporter(s) involved in the delivery of arginine and ADMA into the CNS, may allow us to explain the mechanism behind the arginine paradox, whereby nutritional supplementation with arginine increases NO production despite the intracellular NOS being saturated with arginine [18]. Thus, the observed benefit of arginine supplementation cannot simply be explained by an increase in arginine delivery into the cell, another mechanism of action must be involved.

## Materials and methods

### Materials

[3H]-arginine (mol. wt., 174.2 g/mol; specific activity, 43 Ci/mmol; >97% radiochemical purity) was purchased from Amersham Radiochemicals, Buckinghamshire, UK.[3H]-ADMA (mol. wt., 276.7 g/mol; specific activity, 8 Ci/mmol; 96.4% radiochemical purity) was synthesised and tritiated by Amersham Radiochemicals, Cardiff, UK.

[14C]-sucrose (mol. wt., 342.3 g/mol; specific activity, 0.412 Ci/mmol; 99% radiochemical purity; Moravek Biochemicals, Brea, CA, USA). Unlabelled *L*-arginine and asymmetric dimethylarginine ($N^G,N^G$-dimethylarginine dihydrochloride or ADMA) were purchased from Sigma Aldrich (Dorset, UK). All transport inhibitors used were purchased from Sigma Aldrich; Dorset, UK and were readily soluble in the artificial plasma – requiring no extra steps for dissolution beyond standard mechanical stirring and heating to physiological temperature (37 °C).

### Animals

All experiments requiring the use of animals were performed in accordance with the Animal (Scientific Procedures) Act, 1986 and Amendment Regulations 2012 and with consideration to the Animal Research: Reporting of *In Vivo* Experiments (ARRIVE) guidelines. The study was approved by the King's College London Animal Welfare and Ethical Review Body and performed under license: 70/6634. In the following sections we also describe attempts made to reduce the number of animals used in our studies in line with the principles of the 3Rs (replacement, refinement, and reduction). It is noted that all the raw data was obtained during a period of three years by the same researcher.

All animals used in procedures were adult male BALB/c mice (between 23 g and 25 g) sourced from Harlan Laboratories, Oxon, UK, unless otherwise stated. All animals were maintained under standard temperature and lighting conditions. Access to water and food was provided *ad libitum*. Welfare was monitored daily by animal technicians. Animals which were showing signs of distress (such as not feeding or drinking) were brought to the attention of the named veterinary

surgeon and the animal euthanized. Mice were used for all the procedures under a non-recovery anaesthetic and represent the animals of lowest neurological sensitivity to which the protocols can be successfully applied. Domitor® (medetomidine hydrochloride) and Vetalar® (ketamine) were both purchased from Harlan Laboratories; Cambridge, UK. All animals were anaesthetised (2 mg/kg Domitor® and 150 mg/kg Vetalar® injected intraperitoneally), and a lack of self-righting and paw-withdrawal reflexes (surrogate indicators of consciousness) thoroughly checked prior to carrying out all procedures. Animals were heparinised with 100 units heparin (in 0.9% m/v NaCl(aq), Harlan Laboratories; Oxon, UK), administered *via* the intraperitoneal route prior to surgery.

### *In situ* brain perfusion technique

The *in situ* brain/choroid plexus perfusion method was performed as previously described and will only be briefly detailed here [17,19]. An artificial plasma containing [$^3$H]-arginine (11.6 nM) or [$^3$H]-ADMA (62.5 nM) and [$^{14}$C]-sucrose was infused into the left ventricle of the heart for 10 minutes at a flow rate of 5 mL/min. The artificial plasma consisted of a modified Krebs-Henseleit mammalian Ringer solution with the following constituents: 117 mM NaCl, 4.7 mM KCl, 2.5 mM CaCl$_2$, 1.2 mM MgSO$_4$, 24.8 mM NaHCO$_3$, 1.2 mM KH$_2$PO$_4$, 10 mM glucose, and 1 g/liter bovine serum albumin. At the end of the perfusion period a CSF sample was taken from the cisterna magna, the animal was decapitated, the brain removed and samples taken under a Leica S4E L2 stereomicroscope (Leica; Buckinghamshire, UK). The samples included the frontal cortex, caudate nucleus, occipital cortex, hippocampus, hypothalamus, thalamus, pons and cerebellum. The circumventricular organs (CVOs) including the choroid plexus, pituitary gland and pineal gland were also sampled. Brain samples were also taken for capillary depletion analysis.

[$^{14}$C]-sucrose is a baseline marker molecule. It is a polar, hydrophilic molecule, which does not cross membranes well, and can be used to measure extracellular space including the vascular space. It is specifically used in brain samples (including brain homogenate and brain supernatant) to measure the vascular space at all perfusion time points. At later time points, in particular, it may also represent the capillary endothelial cell volume and/ or diffusion across the brain capillary wall and into the brain interstitial fluid. In the pellet samples it will measure extracellular fluid and may also represent capillary endothelial cell volume. In the choroid plexus samples, [$^{14}$C]-sucrose would represent vascular space, plus the interstitial fluid volume in the compartment between choroid plexus capillary endothelial cells and the choroid plexus epithelial cells of the blood-CSF barrier. It may also represent any accumulation into the cells of the choroid plexus. In the pituitary and pineal gland samples, [$^{14}$C]-sucrose would measure the vascular space plus the interstitial fluid compartment between the blood vessels and the tanycytic barrier. In CSF samples, [$^{14}$C]-sucrose can be used to correct for possible contamination with blood.

**Capillary depletion analysis.** Capillary depletion analysis involved preparing a whole brain homogenate by the addition of a capillary depletion buffer (141 mM NaCl, 4.0 mM KCl, 2.8 mM CaCl$_2$, 1.0 mM MgSO$_4$, 10.9 mM HEPES, 1.0 mM NaH$_2$PO$_4$, 10 mM glucose at pH 7.4) and a dextran (MW 60,000–90,000) solution (final concentration 13%) [17,19]. Both the physiological capillary depletion buffer and dextran solution were maintained at 4 °C to halt any cell metabolism and transport processes, and to reduce any cellular or protein damage due to heat generated from the homogenisation process. The whole brain homogenate was then taken for dextran density centrifugation (5,400 x g) and 4 °C to produce an endothelial cell-enriched pellet and supernatant containing brain parenchyma.

**Liquid scintillation analysis.** All samples (brain regions, brain homogenate, supernatant, pellet, CVOs, CSF and plasma samples) were solubilised in 0.5 mL tissue solubiliser (Solvable; Perkin-Elmer; Boston, MA, USA). After incubating the samples at room temperature for 48 h, 4 mL scintillation fluid (Lumasafe®; Perkin-Elmer; Boston, MA, USA) was added to each before vigorous vortexing. The amount of [$^3$H]- and [$^{14}$C]-radioactivity in each sample were then quantified using a Packard Tri-Carb 2900TR liquid scintillation counter (Perkin-Elmer; Boston, MA, USA). Counts per minute were then converted to disintegrations per minute (dpm) by the counter using internally stored quench curves from standards.

## Experimental design

**Transporter specificity experiments.** The identity of the transporter(s) was studied by measuring the [3H]-arginine or [3H]-ADMA uptake into the CNS in the absence and presence of specific transporter inhibitors in the artificial plasma at a perfusion time of 10 minutes and the results compared. These inhibitors included 20 mM *L*-homoarginine, 4 mM 2-amino-endo-bicyclo[2.2.1]heptane-2-carboxylic acid (BCH), 500 µM a-methyl-*D,L*-tryptophan, 200 µM *L*-phenylalanine, 5 mM *L*-leucine or 2 mM harmaline. The transporter proteins these inhibitors target are listed in S1 Table in S1 File.

It is noted that the data obtained from the [3H]-arginine control group at 10 minutes and the [3H]-ADMA control group at 10 minutes data have recently been published [17]. In this present study, these data sets were combined with new data sets obtained using the transporter inhibitors to identify specific transporter interaction. This allowed us to reduce the number of animals needed for this study in line with the principles of the 3Rs (replacement, refinement, and reduction).

**Self-inhibition experiments.** A series of single-time point experiments involved varying the unlabelled concentration of ADMA in the artificial plasma and measuring the [3H]-ADMA uptake into the brain regions, capillary depletion samples, CVOs and CSF after 10 min of perfusion. Specifically, the uptake of [3H]-ADMA (62.5 nM) and [14C]-sucrose into the CNS was measured in the absence and presence of 0.5, 3.0, 10, 100 and 500 µM un-labelled ADMA in the artificial plasma for a perfusion period of 10 minutes. This enabled calculation of the kinetic constants; $V_{max}$ (the maximal influx rate of the saturable component), $K_m$ (the half-saturation constant) and $K_d$ (the constant of non-saturable diffusion).

These concentrations were selected as the plasma concentration for ADMA under normal physiological conditions is typically found to be approximately 0.5 µM in humans and in several mammalian species, including the mouse [20–23]. Plasma concentrations for ADMA under pathological conditions are typically in the range of 3.0 µM [24,25]. Concentrations above 10 µM can be considered as supraphysiological as ADMA would not ordinarily be found at concentrations that high in the body in health or disease. However, the higher ADMA concentrations can help determine the kinetic characteristics of transport.

It is noted that the data obtained from the [3H]-ADMA group in the presence of 100 µM unlabelled ADMA data has recently been published [17]. In this present study this existing data set was combined with new data sets using unlabelled ADMA at four other concentrations (0.5, 3.0, 10 and 500 µM) in order to calculate previously unreported kinetic constants ($K_m$, $V_{max}$ and $K_d$). This allowed us to reduce the number of animals needed in line with the principles of the 3Rs. Kinetic constants are valuable as they allow data from different assays to be compared.

## Expression of results

**Relative uptake from the perfusate into the tissues.** The concentration of radioactivity in the brain regions ($C_{Brain}$; dpm/g), CVOs ($C_{CVO}$; dpm/g) and CSF ($C_{CSF}$; dpm/ml) is expressed as a percentage of that in the artificial plasma ($C_{pl}$; dpm/ml) and termed $R_{BRAIN}$ (dpm/100g), $R_{CVO}$ (dpm/100g) or $R_{CSF}$ (dpm/100 ml) respectively or $R_{Tissue}$ as shown in equation 1.

$$R_{Tissue} = \frac{C_{Tissue}}{C_{pl}} \times 100$$

(1)

Correcting for vascular space involved subtracting the $R_{Tissue}$ value obtained for [14C]-sucrose in each sample from the $R_{Tissue}$ value concurrently obtained for the [3H]-labelled solute of interest (i.e., [3H]-arginine or [3H]-ADMA).

The percentage change in the $R_{Tissue}$ uptake values achieved in the absence or presence of an unlabelled inhibitor can be determined by means of equation 2.

$$\% \ change = \frac{RTissue - RInhibition}{RTissue} \times 100$$

(2)

where $R_{Inhibition}$ is the $R_{Tissue}$ uptake in the presence of an inhibitor in the artificial plasma. A decrease in the uptake of radiolabelled solute in the presence of unlabelled inhibitor is indicative of a saturable influx transport system. Conversely, an increase in the uptake of radiolabelled solute in the presence of unlabelled inhibitor is indicative of a saturable efflux transport system. No change in the distribution of the radiolabelled solute in the absence and presence of the unlabelled inhibitor may indicate the absence of saturable transport by the radiolabelled solute or the use of influx and efflux transporters by the radiolabelled solute.

**Calculation of kinetic constants.** The rate of transfer across the BBB and the blood-CSF barrier can be calculated as a transfer constant ($K_{in}$) as shown in equation 3 [17].

$$K_{in} = \frac{C_{Tissue}\ (T)}{C_{pl}\ T} \tag{3}$$

where $C_{Tissue}$ (T) is radioactivity (dpm) per g of tissue at time-point T (perfusion time in minutes), and $C_{pl}$ is radioactivity (dpm) per mL of artificial plasma.

Calculating a unidirectional transfer constant by means of equation 3, assumes that the entry of the radiolabelled solute of interest into the CNS is less than, but proportional to, its concentration in the artificial plasma, and CNS to blood efflux is much smaller than blood to CNS influx of the test solute and therefore can be ignored [26].

It should however be noted that calculating the $K_{in}$ value from blood to CNS using this method requires that $R_{Tissue}$ at time T is first corrected for vascular space by subtracting the $R_{Tissue}$ value for [$^{14}$C]-sucrose determined at that time-point.

In the case where the uptake of the radiolabelled test amino acid is examined in the presence of various unlabelled concentrations of the same test solute, $K_{in}$, can be defined as in equation 4:

$$K_{in} = \frac{V_{max}}{K_m + C_{cap}} + K_d \tag{4}$$

where $V_{max}$ is the maximal influx rate of the saturable component; $K_m$ is the half-saturation constant; $K_d$ is the constant of non-saturable diffusion and $C_{cap}$ is the mean capillary concentration of the amino acid.

Under the present experimental conditions, flow to the brain (F) is always greater than $1\,ml.min^{-1}.g^{-1}$, which is much greater than the highest measured $K_{in}$ (i.e., F>> $K_{in}$), the difference between $C_{Cap}$ and concentrations of amino acid in the artificial plasma, $C_{pl}$, becomes negligible and equation 4 can be simplified to equation 5:

$$K_{in} = \frac{V_{max}}{K_m + C_{pl}} + K_d \tag{5}$$

Unidirectional ADMA flux ($J_{in}$) into the brain (nmol.min$^{-1}$.g$^{-1}$) and CSF (nmol.min$^{-1}$.ml$^{-1}$) was determined using equations 6 and 7:

$$J_{in} = F\left(1 - e^{-Kin/F}\right) C_{pl} \tag{6}$$

and since F>> $K_{in}$ equation 6 approximates to:

$$J_{in} \approx K_{in} C_{pl} \tag{7}$$

Unidirectional influx of amino acids ($J_{in}$) into the CNS can be related to the kinetic constants $K_m$, $V_{max}$ and $K_d$ by equation 8 where total flux is equal to the sum of the saturable flux and non-saturable flux:

**TOTAL FLUX $=$ SATURABLE FLUX $+$ NON $-$ SATURABLE FLUX**

$$J_{in} = \frac{V_{max}C_{pl}}{K_m + C_{pl}} + K_d C_{pl}$$

(8)

When appropriate, $K_d$ was determined as the slope of the line from linear regression analysis of the total flux measured at the two highest ADMA concentrations utilized (i.e. 100 and 500 µM). Linear regression analysis of the data was performed using Graph Pad Prism (Version 10.2.2). This constant of non-saturable diffusion was then used to calculate a non-saturable flux which could be subtracted from the total flux to determine the saturable flux. Saturable flux was then plotted as a function of concentration and using Michaelis-Menten Kinetic analysis in Graph Pad Prism (Version 10.2.2) estimates of the best-fit values of $V_{max}$ and $K_m$ obtained. In some cases, there was no measurable non-saturable component and so estimates of $V_{max}$ and $K_m$ were obtained from plots of total flux, which was equal to the saturable flux.

## Statistics

Data from all experiments are presented as mean $\pm$ standard error of the mean (SEM). Results were grouped into brain regions, capillary depletion samples and CVOs and CSF for presentation and data analysis as appropriate. One-way ANOVA (Gaussian distribution assumed and equal SD not assumed) followed by Dunnett's post-hoc test was used to compare the means of the control group to the test group. The results of the post-hoc test were reported. Unpaired two tailed t-tests (Gaussian distribution and equal SD assumed) was used to compare two groups. Statistical significance was taken as follows: not significant (ns; $p > 0.05$) or significant (*$p < 0.05$, **$p < 0.01$, ***$p < 0.001$). Statistical analyses were performed using GraphPad Prism v5.0c or v6 graphing and statistics package for Mac or using GraphPad Prism v10.2.2 for Windows.

## Results

### Arginine

The transport of [³H]-arginine and [¹⁴C]-sucrose across the blood-brain and blood-CSF barriers was examined in the presence of transporter inhibitors. All inhibitors were used at previously published concentrations known to inhibit specific transport systems (S1 Table in S1 File). [¹⁴C]-sucrose distribution into each of the brain regions, capillary depletion samples, CVO and CSF was not significantly affected by the presence of 20 mM *L*-homoarginine, 4 mM BCH or 500 µM α-methyl-*D,L*-tryptophan (S1–S3 Figs in S1 File). This indicates that these inhibitors did not affect the integrity of the blood-brain and blood-CSF barriers in the [³H]-arginine experiments.

The uptake of [³H]-arginine is almost totally inhibited (up to 99.7%) by 20 mM *L*-homoarginine in all brain regions, except the hypothalamus where the reduced uptake of 98.0% did not attain statistical significance due to the wider variability of the values in this region (Fig 1). The inclusion of 4 mM BCH or 500 µM α-methyl-*D, L*-tryptophan in the artificial plasma did not significantly affect the uptake of [³H]-arginine uptake in the majority of brain regions sampled. The exception was the hippocampus sample where the inclusion of 500 µM α-methyl-*D, L*-tryptophan in the artificial plasma significantly inhibited [³H]-arginine uptake by 49.4%.

As you may expect from the data described above, the uptake of [³H]-arginine was also almost totally inhibited (up to 99.7%) by 20 mM *L*-homoarginine in the whole brain homogenate, supernatant containing brain parenchyma, and endothelial cell-enriched pellets following capillary depletion analysis (Fig 2). There was also no significant effect of 4 mM BCH or 500 µM α-methyl-*D,L*-tryptophan on [³H]-arginine distribution in these samples.

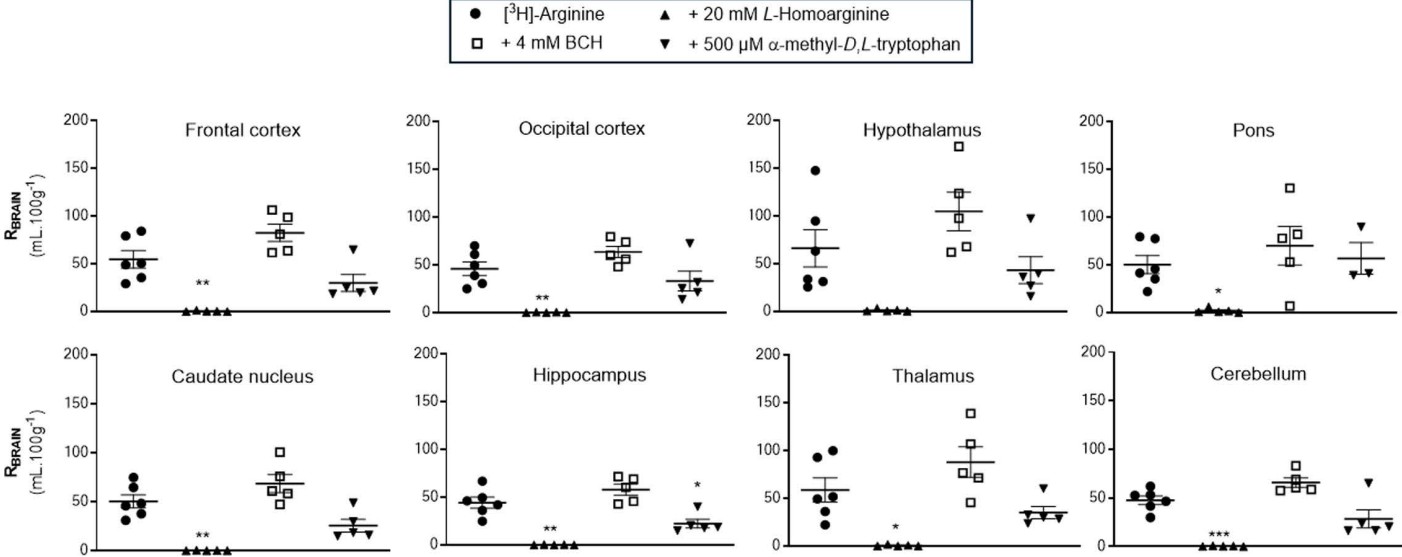

**Fig 1. The effect of L-homoarginine, BCH and α-methyl-D,L-tryptophan on the regional brain uptake of [³H]-arginine (10 minute perfusion).** [³H]-arginine uptake is expressed as the percentage ratio of the [³H]-arginine concentration in the tissue to the [³H]-arginine concentration in the plasma (mL.100 g⁻¹) and is corrected for [¹⁴C]-sucrose (vascular space). Each bar represents the mean ± SEM of 5-6 animals, except pons where it was n of 3 for the α-methyl-*D,L*-tryptophan group. Each marker represents one animal. One-way ANOVA with Dunnett's post-hoc test was used to compare means to control ([³H]-arginine alone), with statistical significance taken as *$p < 0.05$, **$p < 0.01$, ***$p < 0.001$ (GraphPad Prism 10.2 for Windows).

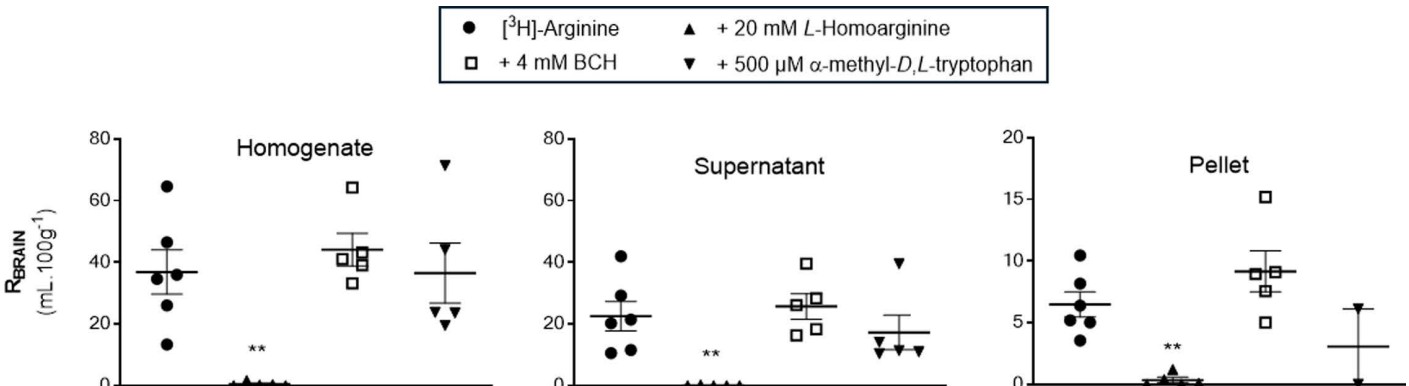

**Fig 2. The effect of L-homoarginine, BCH and α-methyl-D,L-tryptophan on the distribution of [³H]-arginine in capillary depletion samples (10 minute perfusion).** [³H]-arginine uptake is expressed as the percentage ratio of the [³H]-arginine concentration in the tissue to the [³H]-arginine concentration in the plasma (mL.100 g⁻¹) and is corrected for [¹⁴C]-sucrose. Each bar represents the mean ± SEM of 4-6 animals, except the α-methyl-*D,L*-tryptophan pellet group where it was n = 2. Each marker represents one animal. One-way ANOVA with Dunnett's post-hoc test was used to compare means to control ([³H]-arginine alone), with statistical significance taken as *$p < 0.05$, **$p < 0.01$, ***$p < 0.001$ (GraphPad Prism 10.2.2 for Windows).

The distribution of [³H]-arginine into the choroid plexus and pituitary gland was significantly inhibited by 20 mM *L*-homoarginine being 99.8% and 97.6% respectively (Fig 3). Interestingly, although there was a decrease of 88.4% in the distribution of [³H]-arginine in the presence of *L*-homoarginine into the CSF, this failed to attain statistical significance. This will be related to the difficulties in taking CSF samples in these small animals and as a consequence the low sample size of 3 for this inhibitor group (Fig 3). In agreement with observations in all the other samples, (except the hippocampus), the

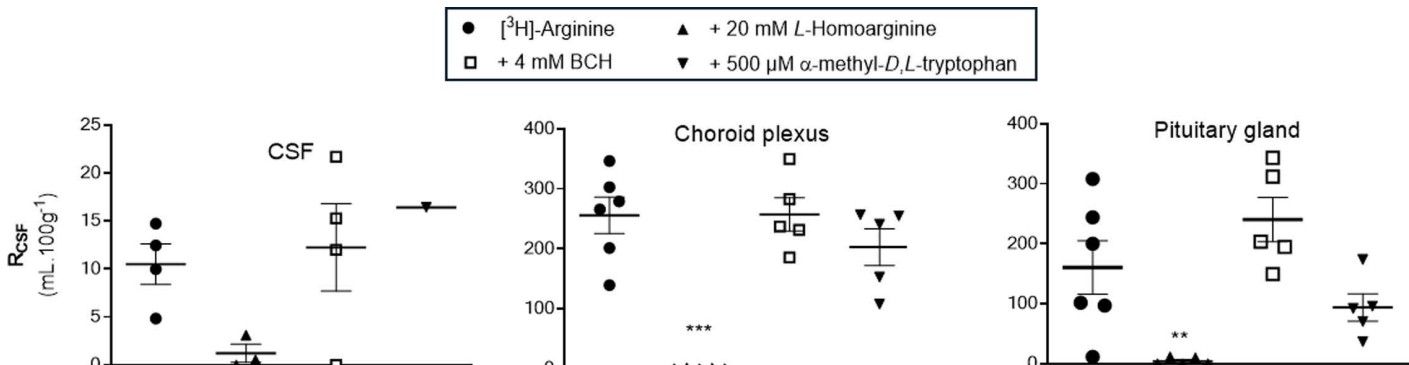

**Fig 3. The effect of L-homoarginine, BCH and α-methyl-D,L-tryptophan on the distribution of [³H]-arginine in CSF, choroid plexus and pituitary gland (10 minute perfusion).** [³H]-arginine uptake is expressed as the percentage ratio of the [³H]-arginine concentration in the tissue or CSF to the [³H]-arginine concentration in the plasma (mL.100 g⁻¹) and is [¹⁴C]-sucrose corrected. Each bar represents the mean ± SEM of 3-6 animals except the α-methyl-D,L-tryptophan CSF group where it was n = 1. Each marker represents one animal. One-way ANOVA with Dunnett's post-hoc test was used to compare means to control ([³H]-arginine alone), with statistical significance taken as *$p < 0.05$, **$p < 0.01$, ***$p < 0.001$ (GraphPad Prism 10.2.2 for Windows).

inclusion of either 4 mM BCH or 500 µM α-methyl-D,L-tryptophan in artificial plasma had no statistically significant effect on the distribution of [³H]-arginine into the CSF, choroid plexus and pituitary gland.

**ADMA.** The transport of [³H]-ADMA and [¹⁴C]-sucrose across the blood-brain and blood-CSF barriers was also examined in the presence of specific transport inhibitors (S1 Table in S1 File). All inhibitors were used at previously published concentrations known to inhibit the specific transport systems of interest. To explore if the inhibitor affected the integrity of the blood-CNS interfaces we first compared the [¹⁴C]-sucrose values in the absence and presence of the various inhibitors in these [³H]-ADMA experiments (S4–S6 Figs in S1 File).

[¹⁴C]-sucrose distribution into each of the brain regions including the capillary depletion analysis compartments and CVOs was not significantly affected by most of the inhibitors, i.e., 20 mM L-homoarginine, 4 mM BCH, 500 µM α-methyl-D,L-tryptophan or 2 mM harmaline (S4–S6 Figs in S1 File). However, two of the inhibitors, 5 mM L-leucine and 200 µM L-phenylalanine did significantly change [¹⁴C]-sucrose distribution in some of the samples. Specifically, the [¹⁴C]-sucrose space was significantly higher than expected with 5 mM L-leucine in the following samples: frontal cortex, caudate nucleus, cerebellum and pellet (S4 and S5 Figs in S1 File). This suggests that L-leucine at a concentration of 5 mM could damage membranes (pellet; S5 Fig in S1 File) resulting in loss of BBB integrity (brain regions, S4 Fig in S1 File). Interestingly, 200 µM L-phenylalanine caused a significant reduction in the vascular space of the occipital cortex and hippocampus possibly as a result of vasoconstriction and/or inadequate perfusion (S4 Fig in S1 File). However, it is noted that the [¹⁴C]-sucrose values achieved with phenylalanine in these two regions are well within the range of [¹⁴C]-sucrose values achieved with the other inhibitor groups (S4 Fig in S1 File), so the difference observed with phenylalanine could be ignored. Although in these experiments a reduction of vascular space is less of a concern than an increase in vascular space, which suggests loss of BBB integrity, the effects of both these inhibitors on the distribution of [³H]-ADMA into these specific samples were not used to draw any conclusions about the transporter sensitivity of [³H]-ADMA. However, all data sets are presented for comparison (Figs 4 and 5; S2 and S3 Tables in S1 File).

The uptake of [³H]-ADMA is almost totally inhibited by 20 mM L-homoarginine in all brain regions (up to 99.2%; Fig 4; S2 Table in S1 File). The inclusion of either 4 mM BCH, 500 µM α-methyl-D,L-tryptophan, 200 µM L-phenylalanine, 5 mM L-leucine, or 2 mM harmaline in artificial plasma also reduced [³H]-ADMA uptake into brain regions, but to a much lesser degree, and only attaining statistical significance in some regions (Fig 4; S2 Table in S1 File). It is noted that the inhibitors, L-leucine and L-phenylalanine, could still significantly inhibit [³H]-ADMA uptake in those brain regions which still had an

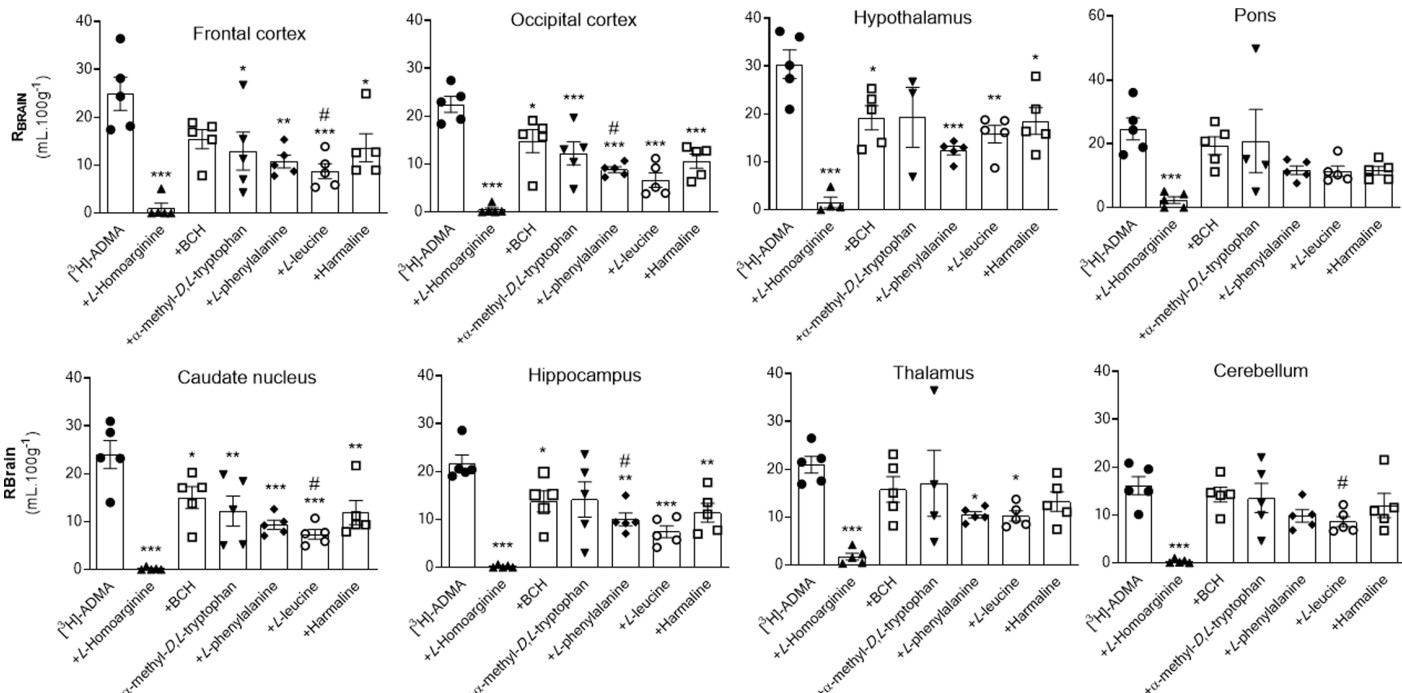

**Fig 4. The effect of 20 mM L-homoarginine, 4 mM BCH, 500 μM α-methyl-D,L-tryptophan, 200 μM L-phenylalanine, 5 mM L-leucine and 2 mM harmaline on the regional brain uptake of [³H]-ADMA (10 minute perfusion).** [³H]-ADMA uptake is expressed as the percentage ratio of the [³H]-ADMA concentration in the tissue to the [³H]-ADMA concentration in the plasma (mL.100 g⁻¹) and is corrected for [¹⁴C]-sucrose (vascular space). Perfusion time is 10 minutes. Each bar represents the mean ± SEM of 4-5 animals. Each marker represents one animal. Asterisks represent one-way ANOVA with Dunnett's post-hoc tests comparing mean±SEM to control within each sample/region, *$p < 0.05$, **$p < 0.01$, ***$p < 0.001$ (GraphPad Prism 6.0 for Mac). #The [¹⁴C]-sucrose (vascular space; S4 Fig in S1 File) in that brain region was statistically affected by the presence of the inhibitor (i.e., *L*-phenylalanine or *L*-leucine).

intact BBB as measured by [¹⁴C]-sucrose (e.g., hypothalamus and thalamus; S2 Table in S1 File). *L*-homoarginine also inhibited uptake of [³H]-ADMA in whole brain homogenate (by 78.9%) and brain parenchyma (supernatant; by 65.8%) following capillary depletion (Fig 5; S3 Table in S1 File). *L*-phenylalanine and *L*-leucine inhibited [³H]-ADMA distribution into whole brain homogenate (by 61.3% and 63.7% respectively), but not brain parenchyma (supernatant) following capillary depletion (Fig 5; S3 Table in S1 File). The inclusion of BCH, α-methyl-*D,L*-tryptophan or harmaline into the artificial plasma did not significantly affect the uptake of [³H]-ADMA into either the brain homogenate or supernatant. None of the inhibitors inhibited uptake of [³H]-ADMA in the endothelial cell enriched pellet (Fig 5; S3 Table in S1 File).

None of the inhibitors affected the distribution of [³H]-ADMA in CSF (Fig 6; S4 Table in S1 File). All the inhibitors inhibited the distribution of [³H]-ADMA in the pineal gland (Fig 6; S4 Table in S1 File). The distribution of [³H]-ADMA in the choroid plexus and pituitary gland was inhibited by all inhibitors except for 500 μM α-methyl-*D,L*-tryptophan (Fig 6; S4 Table in S1 File).

## Kinetic constants

The transport of [³H]-ADMA (62.5 nM) and [¹⁴C]-sucrose across the blood-brain and blood-CSF barriers was examined in the presence of different concentrations of unlabelled ADMA (0.5–500 μM) and this data was used to calculate the kinetic constants, $K_m$, $V_{max}$ and $K_d$. To explore if the unlabelled ADMA affected the integrity of the blood-CNS interfaces we first compared the [¹⁴C]-sucrose values in the absence and presence of the various concentrations of unlabelled ADMA (S7–S9 Figs in S1 File).

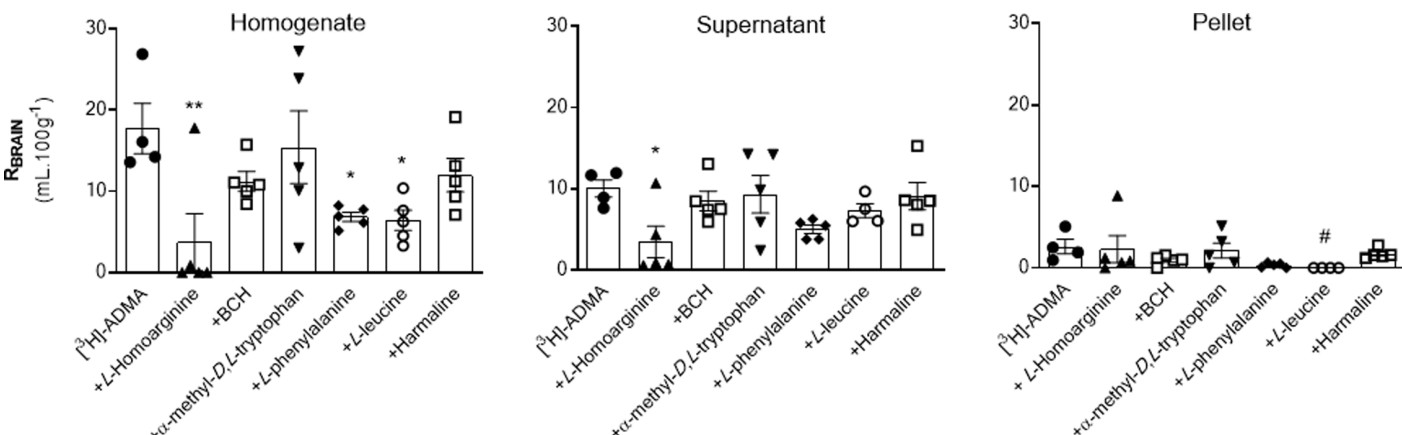

**Fig 5. The effect of 20 mM L-homoarginine, 4 mM BCH, 500 µM α-methyl-D,L-tryptophan, 200 µM L-phenylalanine, 5 mM L-leucine and 2 mM harmaline on the distribution of [³H]-ADMA in capillary depletion samples (10 minute perfusion).** [³H]-ADMA uptake is expressed as the percentage ratio of the [³H]-ADMA concentration in the tissue to the [³H]-ADMA concentration in the plasma (mL.100 g⁻¹) and is corrected for [¹⁴C]-sucrose. Perfusion time is 10 minutes. Each bar represents the mean ± SEM of 4-5 animals. Each marker represents one animal. Asterisks represent one-way ANOVA with Dunnett's post-hoc tests comparing mean±SEM to control within each region/sample, *$p < 0.05$, **$p < 0.01$ (GraphPad Prism 6.0 for Mac). #The [¹⁴C]-sucrose value (S5 Fig in S1 File) in the pellet was statistically affected by the presence of L-leucine.

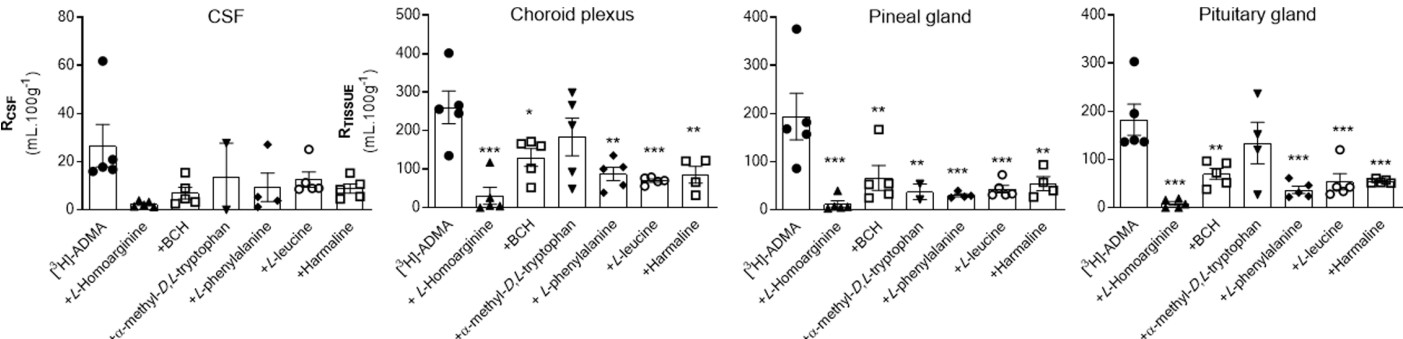

**Fig 6. The effect of 20 mM L-homoarginine, 4 mM BCH, 500 µM α-methyl-D,L-tryptophan, 200 µM L-phenylalanine, 5 mM L-leucine and 2 mM harmaline on the distribution of [³H]-ADMA in CSF, choroid plexus and CVOs (10 minute perfusion).** [³H]-ADMA uptake is expressed as the percentage ratio of the [³H]-ADMA concentration in the tissue or CSF to the [³H]-ADMA concentration in the plasma (mL.100 g⁻¹) and has been corrected for [¹⁴C]sucrose. Perfusion time is 10 minutes. Each bar represents the mean ± SEM of 4-5 animals except for the CSF and pineal sample where with the inhibitor group, α-methyl-D,L-tryptophan n = 2. Each marker represents one animal. Asterisks represent one-way ANOVA with Dunnett's post-hoc tests comparing mean±SEM to control within each region/sample, *$p < 0.05$, **$p < 0.01$, ***$p < 0.001$ (GraphPad Prism 6.0 for Mac).

The [¹⁴C]-sucrose distribution into the brain regions, brain homogenate, brain supernatant, CSF and CVOs in the presence of unlabelled ADMA up to a concentration of 100 µM were not statistically significantly different to the control values achieved in the absence of unlabelled ADMA (S7–S9 Figs in S1 File). The exception to this is the **[¹⁴C]-sucrose** values in the presence of 3 µM unlabelled ADMA, which were statistically different to the control values achieved in the absence of unlabelled ADMA in two brain regions, the hippocampus and thalamus (S7 Fig in S1 File). However, we considered that this difference with 3 µM unlabelled ADMA could be ignored as the values were within the same range as that achieved for [¹⁴C]-sucrose in the presence of the higher unlabelled ADMA concentrations where no statistical difference was observed (i.e., 10 and 100 µM unlabelled ADMA). In contrast, unlabelled ADMA concentrations of 500 µM did statistically increase the distribution of [¹⁴C]-sucrose into the occipital cortex, caudate nucleus, hippocampus, thalamus, pons and cerebellum

and the values measured were at the upper range of [$^{14}$C]-sucrose values achieved at the lower concentrations of unlabelled ADMA (i.e., ≤ 100 μM unlabelled ADMA; S7 Fig in S1 File). As this does suggest loss of BBB integrity in these regions we only interpreted the kinetic characteristics of [$^{3}$H]-ADMA in the other regions where the BBB remained statistically intact (i.e., frontal cortex, hypothalamus, homogenate and supernatant) (S7 and S8 Figs in S1 File). Interestingly, the [$^{14}$C]-sucrose distribution into the pellet samples was statistically reduced by the presence of unlabelled ADMA at most concentrations (S8 Fig in S1 File) and was in the range achieved in other test groups where no statistical difference was obtained (S2 Fig in S1 File). This suggests the membrane was intact in these samples and we could present and interpret the kinetic characteristics of ADMA distribution in this sample as well.

The uptake of [$^{3}$H]-ADMA (62.5 nM) was significantly self-inhibited by 0.5, 3.0, 10, 100 and 500 μM un-labelled ADMA in all brain regions (S10 Fig in S1 File). The uptake of [$^{3}$H]-ADMA into the capillary depletion samples, pineal gland, pituitary gland and choroid plexuses was decreased in the presence of unlabelled ADMA, although it failed to attain statistical significance in the homogenate, supernatant and pellet at a concentration of 10 μM unlabelled ADMA and supernatant at a concentration of 0.5 μM unlabelled ADMA (S11 and S12 Figs in S1 File). Distribution of [$^{3}$H]-ADMA into the CSF was significantly decreased at all concentrations of unlabelled ADMA except the lowest concentration of 0.5 μM, where it failed to reach statistical significance (S12 Fig in S1 File).

The $R_{Tissue}$ uptake data for [$^{3}$H]-ADMA in the absence and presence of unlabelled ADMA into the brain regions, capillary depletion samples and CSF shown in S10–S12 Figs in S1 File, has been [$^{14}$C]-sucrose corrected, and was used to calculate the total flux of [$^{3}$H]-ADMA. The total flux, saturable flux and non-saturable flux of [$^{3}$H]-ADMA into the CNS have been plotted as a function of ADMA plasma concentration and are shown in Figs 7 and 8 and S13 Fig in S1 File. The non-saturable fluxes ($K_d C_{pl}$) have been determined from linear regression analysis as described in the methods. The saturable flux data had been calculated by subtracting the non-saturable flux from the total flux, where appropriate. The total flux of [$^{3}$H]-ADMA into the CSF was equal to the saturable flux and therefore there was no non-saturable component (Fig 8). The $K_m$ and the $V_{max}$ were derived using non-linear regression analysis of the saturable flux data (Enzyme kinetics – Michaelis-Menten, GraphPad Prism 10.0 for Windows). S5 Table in S1 File shows the values for $K_m$, $V_{max}$ and $K_d$. Interestingly, the $K_m$, $V_{max}$ and $K_d$ values in each brain region are similar, even if there was loss of BBB integrity in that region as measured by [$^{14}$C]-sucrose (S13 Fig; S5 Table in S1 File).

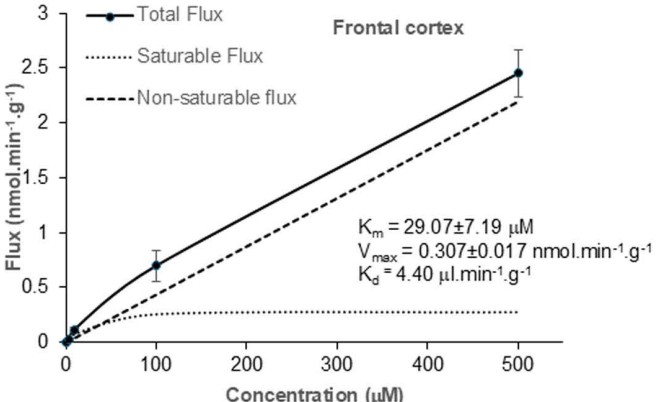 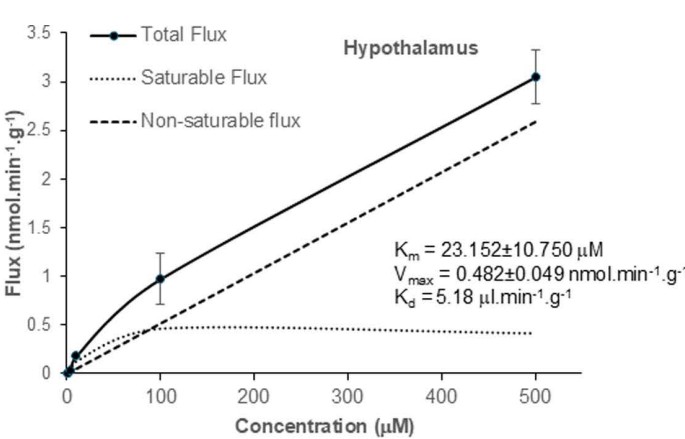

**Fig 7. The contributions of the saturable and non-saturable components to total brain flux of [$^{3}$H]-ADMA are plotted against the unlabelled ADMA concentration.** The measured values are the mean ±SEM for 4-5 mice at each of the 6 ADMA concentrations and has been [$^{14}$C]-sucrose corrected. The lower lines show the contributions of the saturable and non-saturable components to total influx. The $K_d$ value was calculated from linear regression analysis of the total flux at the highest concentrations at 100 and 500 μM. The $K_m$ and $V_{max}$ were calculated by Michaelis-Menten kinetic analysis of the saturable flux (mean values were used). Analyses were performed using GraphPad Prism version 10.

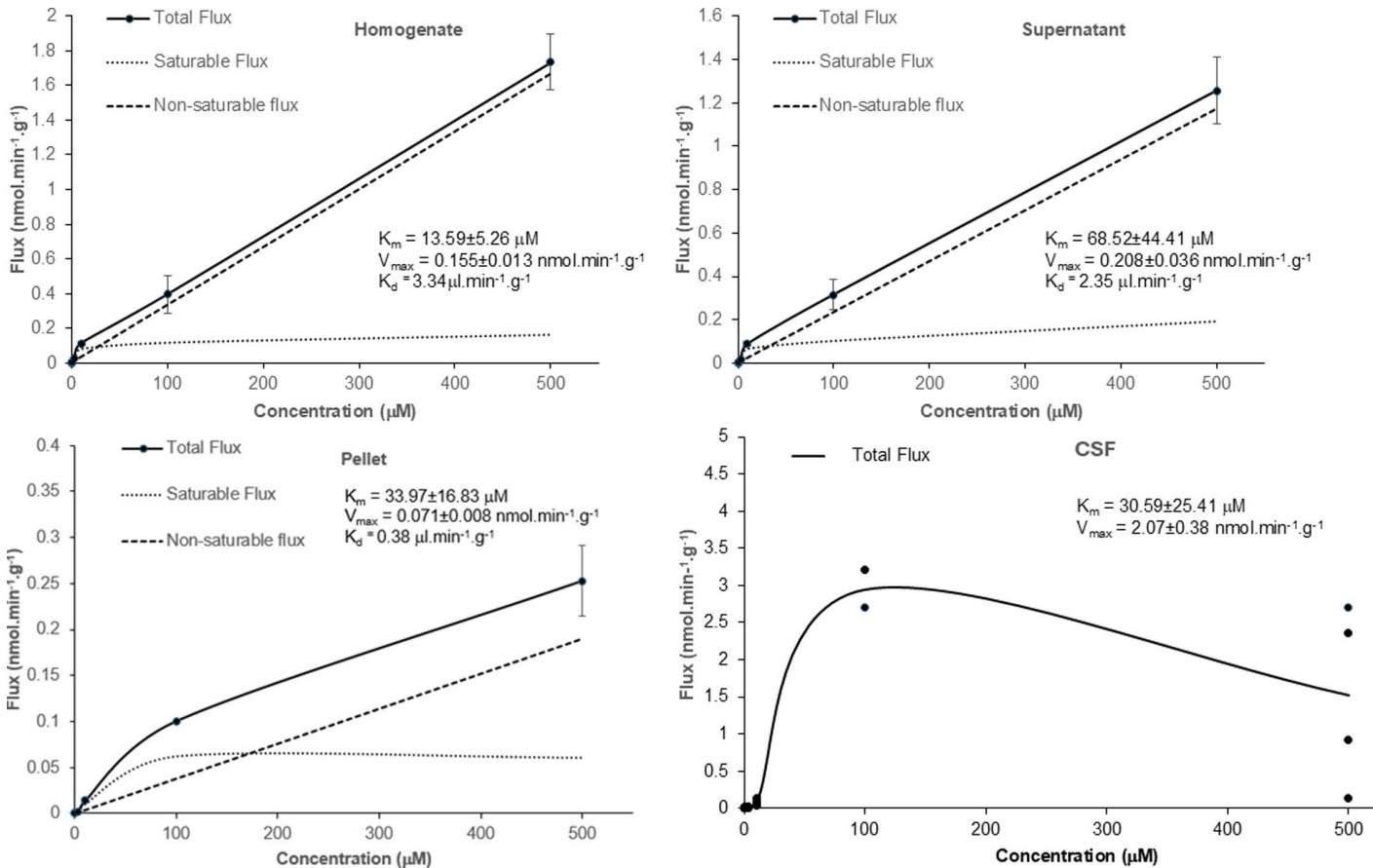

**Fig 8. The contributions of the saturable and non-saturable components to total flux of [³H]-ADMA into the capillary depletion samples are plotted against the unlabelled ADMA concentration.** The measured values are the mean ±SEM for 4-5 mice (homogenate), 4-5 mice (supernatant) and 5 mice (pellet except at 100 µM where it was 1 mouse) at each of the 6 ADMA concentrations and has been [¹⁴C]-sucrose corrected. The lower lines show the contributions of the saturable and non-saturable components to total influx. The $K_d$ value was calculated from linear regression analysis of the total flux at 100 and 500 µM. The $K_m$ and $V_{max}$ were calculated by Michaelis-Menten kinetic analysis of the saturable flux (mean values were used). Analyses were performed using GraphPad Prism version10. **The total flux of [³H]-ADMA into the CSF is plotted against the unlabelled ADMA concentration. Individual values are presented. There was 4-5 mice (except at 100 µM where it was 2 mice) at each of the 6 ADMA concentrations.** The $K_m$ and $V_{max}$ were calculated by Michaelis-Menten kinetic analysis of the total flux (individual values were used) which in this case is the same as saturable flux (GraphPad Prism version10). Unlabelled ADMA did statistically decrease the distribution of [¹⁴C]-sucrose into the pellet (S8 Fig in S1 File). This suggests that the membrane integrity has not been compromised in these samples and could be presented here for comparison.

As the [³H]-ADMA concentration in the choroid plexuses, pituitary gland and pineal gland was higher than the concentration in the plasma at 10 minutes (i.e., $R_{Tissue}$ > 100%; S12 Fig in S1 File), single time uptake analysis could not be used to calculate the $K_{in}$ needed to determine the total flux as described in the methods. Therefore, the $K_m$, $V_{max}$ and $K_d$ values were not determined for these specific samples.

### Transport across the BBB versus the blood-CSF barrier

Statistical analysis using unpaired two tailed t-tests confirmed that at 10 minutes the uptake of [³H]-arginine into the frontal cortex (54.65±9.21 ml.100g⁻¹) was 5.7 times higher than that into the CSF (9.53±2.41 ml.100g⁻¹; P<0.001) and that the uptake of [³H]-arginine into the capillary endothelial cell enriched pellet (6.47±1.018 ml.100g⁻¹) was 39.5 times lower than that into the choroid plexus (255.86±30.41 ml.100g⁻¹; P<0.0001) (S14 Fig in S1 File). All values were corrected for [¹⁴C]-sucrose.

Statistical analysis using unpaired two tailed t-tests confirmed that at 10 minutes the uptake of [3H]-ADMA into the frontal cortex (24.94 ± 3.50 ml.100g$^{-1}$) was not significantly different to that into the CSF (26.95 ± 9.02 ml.100g$^{-1}$; P > 0.05) and that the uptake of [3H]-ADMA into the capillary endothelial cell enriched pellet (2.10 ± 0.86 ml.100g$^{-1}$) was 124.2 times lower than that into the choroid plexus (260.85 ± 42.45 ml.100g$^{-1}$; P < 0.001) (S14 Fig in S1 File). All values were corrected for [14C]-sucrose.

[14C]-sucrose distribution into the frontal cortex, CSF, capillary depletion enriched pellet samples and the choroid plexus samples was pooled together from the [3H]-arginine and [3H]-ADMA perfusions and the values compared using unpaired two tailed t-tests (S15 Fig in S1 File). The distribution of [14C]-sucrose into the frontal cortex (3.06 ± 0.38 ml.100g$^{-1}$) was not significantly different to that measured in the CSF (1.75 ± 0.64 ml.100g$^{-1}$). The distribution of [14C]-sucrose into the capillary endothelial cells (0.92 ± 0.28 ml.100g$^{-1}$) was 31.4 times lower than that into the choroid plexus (28.88 ± 4.63 ml.100g$^{-1}$; P < 0.001).

## Discussion

We have previously demonstrated the presence of a saturable transport mechanism(s) at both the blood-brain and blood-CSF barriers for [3H]-arginine and [3H]-ADMA by means of the *in situ* brain/choroid plexus perfusion method in anaesthetized mice [17]. In this present study we investigated the identity of the transporters involved in this saturable transport of [3H]-arginine and [3H]-ADMA using the same method. To do this we included established inhibitors of specific amino acid transporters in the artificial plasma and compared results to those obtained in the absence of inhibitors (S1 Table in S1 File). Transport systems for CAAs, which are expressed at the BBB include: y$^+$, y$^+$L, B$^{0,+}$ and b$^{0,+}$ systems [4,16,27–33]. The term 'transport system' calls attention to the concept that a complex of different proteins rather than a single carrier protein mediates a distinct transport activity (S1 Table in S1 File). Importantly, individual CAA may use more than one transport system.

We first confirmed integrity of the blood-brain and blood-CSF barriers after exposure to the transporter inhibitors by checking that the values achieved for [14C]-sucrose were not significantly different in the absence and presence of the inhibitors (S1–S6 Figs in S1 File). We then progressed to assessing the effect of inhibitors on the CNS delivery of [3H]-arginine and [3H]-ADMA. It was assumed that a significant change in the accumulation of [3H]-arginine or [3H]-ADMA into any CNS region was a result of a transporter interaction. A decrease in accumulation indicating the presence of an influx transporter and an increase in accumulation indicating the presence of an efflux transporter. Importantly, we noted that an absence of an inhibitor effect did not necessarily indicate that there was no significant transporter interaction. This is because transporter involvement in CAAs delivery to the CNS is difficult to interpret conclusively as: (i) several transport systems for CAA are expressed at the BBB and blood-CSF barrier (ii) CAA can be transported by both influx and efflux transporters plus some of the CAA transporters are bidirectional (e.g., system y$^+$) (iii) CAA transporters are expressed on the luminal and/or abluminal membranes of the brain capillary endothelial/choroid plexus epithelial cells and (iv) [3H]-ADMA has been shown to be removed from the CNS [17].

In this present study we found that [3H]-arginine transport into all CNS regions (except CSF) was significantly inhibited (up to 99.7%) by *L*-homoarginine. This indicates that [3H]-arginine is entering into the CNS by system y$^+$. This is because *L*-homoarginine is a substrate for system y$^+$ [34] and the associated transporter proteins, i.e., the cationic amino acid transporter-1 (CAT1; SLC7A1), CAT2A (SLC7A2) and CAT2B (SLC7A3) [35]. In addition, CAT1 is expressed at the BBB and the choroid plexus of the blood-CSF barrier [28,36–38]. Functionally, CAT1 is a bidirectional uniporter which facilitates the movement of CAAs down a concentration gradient and is usually involved in loading cells with amino acids [39]. Interestingly, system-y$^+$ exhibits trans-stimulation, whereby transport is faster when substrate is present on the opposite (*trans-*) side of the membrane [40,41]. System y$^+$ activity has been demonstrated on both the luminal and abluminal plasma membranes of the bovine cerebral capillary endothelium [30]. Interestingly, our data suggests that system-y$^+$ was present (at least) on the blood-side of the cerebral capillary endothelial cells and the choroid plexus due to reduced transport of

[$^3$H]-arginine into the cerebral capillary endothelial cell enriched pellet and the choroid plexus samples in the presence of *L*-homoarginine.

BCH and α-methyl-*D,L*-tryptophan are system B$^{0,+}$ inhibitors, which can inhibit CAA (arginine) and neutral amino acid (leucine) transport by the system B$^{0,+}$ transporter protein, ATB$^{0,+}$ (SLC6A14) when expressed in xenopus oocytes or breast cancer cells [42–44]. However, our *in situ* brain perfusion studies revealed that [$^3$H]-arginine transport into the CNS was not affected by either of the system B$^{0,+}$ inhibitors, BCH or α-methyl-*D,L*-tryptophan. Although it is noted that in one of the 15 CNS regions (i.e., hippocampus) there was a decrease in accumulation (i.e., 47.5%) observed with one (but not both) of the inhibitors (i.e., α-methyl-*D,L*-tryptophan). It is possible that this is just a sampling error but may reflect specific inhibitor sensitivity in this region.

Overall, our data provides evidence that [$^3$H]-arginine predominately uses system-y$^+$, and not system-B$^{0,+}$, to cross both the BBB and the blood-CSF barrier (choroid plexus). This agrees with earlier studies by us and others which showed that: i) transporters for [$^3$H]-arginine could be detected on the luminal membrane of the cerebral capillary endothelium and the blood-side of the choroid plexus [17,30]. ii) arginine can inhibit the cellular uptake of *L*-homoarginine by system-y$^+$ transporter proteins such as CAT1 [35]. iii) [$^3$H]-arginine transport at the human BBB *in vitro* was not affected by the system B$^{0,+}$ substrate, leucine [16]. iv) arginine transport by the transporter protein for system B$^{0,+}$, ATB$^{0,+}$, may not be detectable due to a preference for system y$^+$ and/or the presence of other system B$^{0,+}$ substrates [45] and/or the low expression of ATB$^{0,+}$ in normal tissues [44]. v) the transporter for arginine at the blood-side of the isolated perfused choroid plexus was found not to be sensitive to the system-B$^{0,+}$ inhibitor, BCH [46] and vi) sodium-independent influx of arginine at physiological concentrations occurs predominately by a single and saturable transport system [34].

Our data would support the suggestion that CAT1 provides at least some of the arginine required for NO synthesis [18,47] and has an intermediary role in modulating vascular tone and blood flow.

The next part of our study was to investigate the identity of the transporters involved in the saturable [$^3$H]-ADMA transport across the BBB and the blood-CSF barriers [17]. The significant inhibition of [$^3$H]-ADMA uptake into all the brain regions sampled (up to 99.2%) and the choroid plexus (88.2%) in the presence of 20 mM *L*-homoarginine would suggest that [$^3$H]-ADMA transport at the BBB and blood-CSF barrier can be mediated by system-y$^+$ (Figs 4 and 6; S2 and S4 Tables in S1 File). The transport of [$^3$H]-ADMA into the circumventricular organs (the pineal and pituitary glands) was also sensitive to *L*-homoarginine indicating system y$^+$-involvement (Fig 6). Other studies have also provided evidence that ADMA transport may be sensitive to *L*-homoarginine, as ADMA inhibited *L*-homoarginine uptake into human embryonic kidney (HEK293) cell lines stably overexpressing CAT1 [35]. In addition, we have previously shown that [$^3$H]-arginine transport across the BBB and blood-CSF barrier is inhibited by ADMA and *vice versa* [17] and in this present study we have shown that [$^3$H]-arginine appears to use system y$^+$. Together this information supports a conclusion that [$^3$H]-ADMA transport across the BBB and the blood-CSF barrier involves system-y$^+$ [17].

However, as the uptake of [$^3$H]-ADMA into certain brain regions and the choroid plexus was also significantly inhibited (albeit to a lesser degree) by the presence of the other transporter inhibitors we have evidence that other transporter systems are involved, in addition to system y$^+$ (S2 and S4 Tables in S1 File). Interestingly, our results indicated that [$^3$H]-ADMA uptake into the brain and choroid plexus was inhibited up to 50.0% by the two system B$^{0,+}$ inhibitors (BCH and α-methyl-*D,L*-tryptophan). Therefore, ADMA is likely to be also interacting with system B$^{0,+}$ for transport from plasma to the brain via the BBB and plasma to CSF via the choroid plexus. This is also supported by the fact that [$^3$H]-ADMA delivery to the CNS was inhibited by leucine. Leucine is a neutral amino acid which is not transported by system-y$^+$ [48], but is a substrate for system B$^{0,+}$. In addition, NOS inhibitors, which, like ADMA, are structurally related to arginine (for example N$^G$-nitro-L-arginine), have also been shown to be transported by ATB$^{0,+}$ [49]. We, and others, have previously identified the presence of ATB$^{0,+}$ in human BBB cells (hCMEC/D3) using Western blotting and immunofluorescence [16,27,31]. It is thought to be a sodium-dependent symporter involved in the uptake of amino acids into BBB cells (hCMEC/D3) [16,27,31,39]. Together these facts provide further support that ADMA could also be transported by system B$^{0,+}$ (ATB$^{0,+}$)

across the brain capillary endothelial cells and choroid plexus epithelium. Although, it is noted that ATB$^{0,+}$ mRNA has not been detected in human choroid plexus samples [50].

Interestingly, as well as being transported by ATB$^{0,+}$, leucine and BCH are substrates for system-$L$ [51]. System $L$-amino acid transporters such as the large neutral amino acid transporter 1 protein (LAT1) are encoded by the SLC7A5 gene and forms a heterodimer with the glycoprotein CD98, which is encoded by the SLC3A2 gene. System-$L$ transporters have been shown to be expressed on both luminal and abluminal membranes of the cerebral endothelial cell and the basolateral surface (blood-side) and apical side (CSF-side) of the choroid plexus [4,33,52–55]. LAT1 is a sodium-independent high-affinity transporter for many of the neutral amino acids and prefers those with a bulky side chain, such as leucine and phenylalanine. Both leucine and phenylalanine transport at the choroid plexus has been shown to be sensitive to BCH [52]. Interestingly, the $L$-system carrier at the choroid plexus is thought to be involved in the transport of leucine and phenylalanine from blood to CSF and CSF to blood [54]. Interestingly, our data also show that [³H]-ADMA transport into some regions of the brain and the choroid plexus are also sensitive to phenylalanine with uptake being inhibited by up to 66.4%. Thus as ADMA is positively charged and will not be being transported by the neutral amino acid transporter, system $L$, our data suggests that phenylalanine is interacting with system B$^{0,+}$. Our data would confirm that system B$^{0,+}$ has a broad substrate selectivity (denoted by 'B') accepting neutral (denoted by '0') and cationic (denoted by '+') amino acids as substrates.

Other CAA transport systems may also be involved in [³H]-ADMA delivery to the CNS. For example, system y$^+$L and system b$^{0,+}$. System y$^+$L is a heterodimer formed by the interaction of the heavy chain of the cell surface antigen 4F2 (4F2hc, encoded by the SLC3A2 gene) with the light chains 4F2-lc2 (or y$^+$LAT-1, encoded by SLC7A7) or 4F2-lc3 (or y$^+$LAT-2, encoded by SLC7A6). System y$^+$L transports CAAs with no need for extracellular Na$^+$; however, it can transport both small and large neutral amino acids with high affinity in the presence of this cation [56]. In fact, system y$^+$L transporters such as y$^+$LAT1 and y$^+$LAT2 are thought to exchange a neutral amino acid coupled to a sodium ion for a CAA. System y$^+$L transporters are expressed at the human BBB e,g. y$^+$LAT1 (SLC7A7) and y$^+$LAT2 (SLC7A6) [4]. SLC7A7, SLC7A6 and SLC3A2 mRNA has also been detected in human choroid plexus [37,57–59]. In the presence of a CAA loader, CAT1, y$^+$LAT1 mediates efflux of CAAs in exchange for neutral amino acids and so these amino acid transporters exhibit functional co-operation. The fact that leucine and phenylalanine inhibited [³H]-ADMA accumulation could also be explained by stimulation of ADMA efflux via system y$^+$L by these neutral amino acids. As system y$^+$L removes CAA from cells, this transporter may be involved in the removal of ADMA from the CNS. Removal of ADMA from the CNS has previously been observed by our group [17].

System b$^{0,+}$ is a facilitated transporter of CAAs, which is inhibited by the organic cation, harmaline [30]. It transports cationic and zwitterionic amino acids. The transporter protein associated with system b$^{0,+}$ is b$^{0,+}$AT and is encoded by the SLC7A9 gene. b$^{0,+}$AT mRNA is expressed in the human brain capillary endothelial cells, but at low levels [4] and b$^{0,+}$AT mRNA was not found in human choroid plexus samples [60]. However, harmaline significantly inhibited [³H]-ADMA uptake into the brain and choroid plexus suggesting the involvement of system b$^{0,+}$ in ADMA transport across the mouse BBB and blood-CSF interfaces. The conflicting results may reflect species differences in transporter expression and/or lack of inhibitor specificity for the system b$^{0,+}$ transporter.

Interestingly, [³H]-ADMA accumulation into the brain regions was more sensitive to the transporter inhibitor, L-homoarginine, than unlabelled ADMA (Fig 4; S10 Fig; S2 Table in S1 File). For example, 20 mM L-homoarginine decreased the uptake of [³H]-ADMA by 95.7%, whereas 500 µM unlabelled ADMA only decreased the uptake of [³H]-ADMA by 79.9%, into the frontal cortex. This could be related to concentration differences, but may be because [³H]-ADMA delivery to the CNS involves multiple cationic amino acid transporters such as system y$^+$, B$^{0,+}$, y$^+$L and b$^{0,+}$. As these transporters are involved in the influx and efflux of ADMA, all could be affected to some degree by the presence of unlabelled ADMA and as a result the overall uptake into the brain of ADMA in the presence of unlabelled ADMA could be smaller than that achieved with the specific inhibitor, L-homoarginine.

In this present study we also determined the $K_m$ (half-saturation constant), $V_{max}$ (maximal influx rate of the saturable component) and $K_d$ (diffusion constant) for [$^3$H]-ADMA transport into the different CNS regions using an ADMA concentration range of 62.5 nM – 500 μM at a single time point of 10 minutes (S5 Table in S1 File). For [$^3$H]-ADMA delivery into the frontal cortex the $K_m$, $V_{max}$ and $K_d$ were 29.07 ± 7.19 μM, 0.307 ± 0.017 nmol.min$^{-1}$.g$^{-1}$ and 4.40 μl.min$^{-1}$.g$^{-1}$ respectively. This compares with [$^3$H]-ADMA delivery into the brain homogenate having a $K_m$ of 13.59 ± 5.26 μM, $V_{max}$ of 0.155 ± 0.013 nmol.min$^{-1}$.g$^{-1}$ and $K_d$ of 3.34 μl.min$^{-1}$.g$^{-1}$, into the brain supernatant having a $K_m$ of 68.52 ± 44.41 μM, $V_{max}$ of 0.208 ± 0.036 nmol.min$^{-1}$.g$^{-1}$ and $K_d$ of 2.35 μl.min$^{-1}$.g$^{-1}$ and into the pellet of a $K_m$ of 33.97 ± 16.83 μM, $V_{max}$ of 0.071 ± 0.008 nmol.min$^{-1}$.g$^{-1}$ and $K_d$ of 0.38 μl.min$^{-1}$.g$^{-1}$. The $K_m$ and $V_{max}$ values for [$^3$H]-ADMA accumulation into the CSF were 30.59 ± 25.41 μM, and 2.07 ± 0.38 nmol.min$^{-1}$.g$^{-1}$, respectively. We could not detect a non-saturable component to the total flux for [$^3$H]-ADMA delivery into the CSF, so were not able to calculate a diffusion constant ($K_d$) for this region. As we have evidence that ADMA uses multiple transporters to cross the BBB and blood-CSF barrier and can cross from blood-to-CNS and CNS to blood, it is not possible to assign the kinetic constants to a specific transporter.

The fact that several CAA transport systems are able to transport ADMA across plasma membranes suggests that its intracellular concentration is flexibly regulated within set limits by multiple pathways [61]. Select transport pathways may be active or inactive simultaneously allowing cells to adapt to changing local conditions. Other mechanisms, such as ADMA catabolism via dimethylarginine dimethylaminohydrolase, may also affect intracellular ADMA concentration [62]. Multiple pathways may be expected due to the importance of ADMA to NO bioavailability [25,63]. Plasma ADMA concentrations are significantly raised when compared to control values in Alzheimer's (0.65 ± 0.11 vs 0.54 ± 0.11 μM) [63], ischemic stroke (0.461 ± 0.076 vs 0.433 ± 0.056 μM) [64], type 2 diabetes mellitus (1.59 ± 0.22 vs 0.69 ± 0.04 μM) [65] and hypertension (0.23 ± 0.03 vs 0.10 ± 0.01 μM) [64–66] reaching approximately 3 μM in certain disease states such as peripheral arterial occlusive disease [25]. Interestingly, it is also significantly elevated after prolonged, strenuous exercise (0.37 ± 0.05 vs 0.32 ± 0.05 μM) possibly as a result of exercise-induced muscle damage and inflammation [67]. Elevated ADMA plasma concentrations are associated with diminished NOS activity [25]. The plasma concentration of ADMA in mice has been reported as ranging from 1.07 to 1.58 μM [68]. This suggests that the saturable transport system(s) for [$^3$H]-ADMA delivery into the CNS identified in this study (and having a $K_m$ ranging from 13.6 to 68.5 μM) would not be fully saturated by ADMA in the plasma in health or in disease (S5 Table in S1 File). This may be as expected as a single CAA transport system will transport several CAAs.

Interestingly, our previous study revealed that ADMA and arginine shared a transporter to some extent [17] and the transporter inhibitor studies described in this present study confirmed that arginine and ADMA are transported by system y$^+$. It is interesting to note that system y$^+$ usually binds substrates with a relatively high affinity ($K_m$ < 200 μM) [34] and CAT-1 transports ADMA with a $K_m$ of 183 ± 21 μM [69]. In addition, system y$^+$ is a bidirectional uniporter mediating the influx and efflux of positively charged amino acids with the $K_m$ for influx being lower than the $K_m$ for efflux. This asymmetry may arise in part from the transmembrane potential [34]. The transporter kinetics (S5 Table in S1 File) and plasma concentrations described in this present study and our previous observations [17] would allow arginine supplementation to increase NO production by reducing the intracellular concentration of ADMA either by competitively inhibiting ADMA influx and/or by stimulating ADMA efflux. As the intracellular concentration of ADMA is higher (e.g., 3.6 ± 1.0 μM rat) than the extracellular concentration (0.33 ± 0.10 μM rat) [68] and the y$^+$-system operates by moving cationic amino acids down their concentration gradient so in this case from inside to the outside of cells. It is likely that arginine supplementation would increase NO production by trans-stimulating ADMA efflux rather than inhibiting ADMA influx. Thus, transporter interaction causing a reduction in the intracellular ADMA concentration could explain the arginine paradox, whereby nutritional supplementation with arginine increases NO production despite the intracellular NOS being saturated with arginine. Although other mechanisms possibly in addition to transporter interaction, such as changes to ADMA catabolism [62] and NOS affinity [25,70], cannot be ruled out. Interestingly, normally ADMA concentrations within cells will only inhibit NO production by 10%, but this can increase to 70% when plasma ADMA concentrations are

higher than normal [61]. Therefore, arginine supplementation may only be significantly beneficial when ADMA concentrations are elevated above the normal range.

The advantages and limits of the *in situ* brain perfusion method have recently been described by us [17]. In this present study a further advantage is demonstrated and that is the ability to manipulate the composition of the artificial plasma to measure the kinetic constants ($K_m$, $V_{max}$ and $K_d$) of test molecules (in this case ADMA) transport. A limit of this is that these constants do not necessarily reflect those of a single transporter, but represent blood-CNS (and possibly CNS-blood) transport of the molecule, which could be the result of multiple transporter interactions. A confounding factor of the present study was the actual significance of the statistical difference observed with [$^{14}$C]-sucrose between the control and some of the test groups. In particular when the [$^{14}$C]-sucrose values overlapped with other groups where no statistical difference was detected. We evaluated the relevance of these differences carefully and on a case-by-case level. Importantly, if we remove these groups of data our conclusions still remain the same.

In this study, we also compared the uptake of [$^{3}$H]-arginine and [$^{3}$H]-ADMA into the brain with that into the CSF at 10 minutes (S14 and S15 Figs in S1 File). Interestingly, the uptake of [$^{3}$H]-arginine into the brain was significantly higher than that into the CSF, but the uptake of [$^{3}$H]-ADMA into the brain was not significantly different to that measured in the CSF. In both cases, the accumulation of the radiolabelled test molecule in the choroid plexus was significantly higher than that measured in the cerebral capillary endothelial cell (S14 and S15 Figs in S1 File).

In *in situ* brain/choroid plexus perfusion studies it is not possible to completely separate uptake across the BBB from that across the blood-CSF barrier, thus the relative contributions of the two routes to the concentration of [$^{3}$H]-arginine or [$^{3}$H]-ADMA in the brain or the CSF is not exactly known. In addition, the situation is complicated by the BBB having a lower permeability and greater surface area than the blood-CSF barrier and the complexity of molecule exchange between the brain interstitial fluid and the CSF and *vice versa* [71–73]. However, consideration of the CSF turnover time in rats and humans of approximately 2 and 8 hours, respectively [74,75], and the 10 minute time frame of this mouse perfusion study, does allow us to assume that the (re-)circulating CSF in the glymphatic pathway [1,76] is not the source of the test molecules in the brain parenchyma in these experiments.

Examination of the [$^{3}$H]-arginine, [$^{3}$H]-ADMA and [$^{14}$C]-sucrose accumulation in the different brain regions and compartments, allows us to conclude that [$^{3}$H]-arginine and [$^{3}$H]-ADMA are able to cross the BBB and reach the brain and from there possibly diffuse into the CSF. In addition, direct entry of [$^{3}$H]-arginine and [$^{3}$H]-ADMA into the CSF via the blood-CSF barrier cannot be ruled out. However, the blood-CSF barrier route alone would be unlikely to cause the observed higher concentration of [$^{3}$H]-arginine in the brain than the CSF or the observed homogeneous [$^{3}$H]-ADMA activity between the brain and CSF. This conclusion is reached because the diffusion kinetics from the ventricular CSF into the brain, is considered slower than the rate of bulk flow of CSF out of the ventricles [77]. Thus, the CSF is more likely to act as a 'sink' to the brain than the brain acting as a 'sink' for the ventricular CSF. In addition, even if CSF concentrations exceed brain ISF concentrations and diffusion occurs into the brain ISF, high solute levels may be produced at the ependymal-CSF surface, but as you would expect from a diffusion driven process, there will be an exponential decrease in tissue concentrations at distances as small as a few millimetres from the ventricular surface [77]. Neural cell accumulation would also contribute to this decrease in tissue concentrations. We have previously shown that there was no difference in the distribution of [$^{3}$H]-arginine or [$^{3}$H]-ADMA into the different brain regions including frontal cortex, occipital cortex, hypothalamus, pons, caudate nucleus, hippocampus, thalamus and cerebellum [17]. To conclude, molecules can move from CSF to brain ISF, but compared to the movement across the BBB where diffusional distances between brain capillaries and thus the brain cells is very small, it is very inefficient for reaching large volumes of brain quickly.

The fact that [$^{3}$H]-ADMA has a similar concentration in the brain and CSF, whereas [$^{3}$H]-arginine has a higher concentration in the brain than the CSF, at the 10 minute perfusion time point, may reflect the removal of [$^{3}$H]-ADMA from the brain tissue into the CSF. In fact, brain to CSF flux and subsequent choroid plexus to blood efflux and/or CSF drainage

may contribute to the CNS efflux of ADMA observed in our earlier study [17]. These differences in the handling of [³H]-arginine (mol. wt 174.2 g/mol) and the more lipophilic, [³H]-ADMA (mol. wt. 202.26 g/mol), by the blood-CNS interfaces must be related to differences in brain cell accumulation and/or transporter interaction, which we observed in our present study.

There is substantial evidence that indicates that drugs use transporters to get into and out of cells rather than rely on passive diffusion to cross the lipid bilayer of the plasma membrane [78,79]. For example, system $y^+$ is thought to be able to transport the CAA analogue, eflornithine, which is an anti-human African trypanosomiasis drug [31], and system $B^{0,+}$ is thought to transport the cationic anti-influenza compounds, amantadine and rimantadine [27], across the BBB. Prodrugs of non-steroidal anti-inflammatory drugs have also been successfully developed that exploit the endogenous substrate of the LAT transporter to efficiently deliver the prodrug across the BBB [80]. Interestingly, it has recently been suggested that CAT1 may transport large molecules across the BBB [81] and it may be that AAPB and its metabolites (including exogenous arginine) can enter the brain using this transporter. Another important point is that some cancer-causing viruses also use CAA transporters to get into the brain [82,83]. As mentioned previously, molecules (including drugs) are able to cross the BBB and/or the blood-CSF barrier, but would be unlikely to reach deep brain sites if CNS delivery is solely via the blood-CSF barrier.

## Conclusion

The results of our present study indicate that [³H]-arginine transport at the BBB and the blood-CSF barrier (choroid plexus) involves the cationic amino acid transport system, system $y^+$, but not system $B^{0,+}$. Our study also provides new information that [³H]-ADMA delivery to the CNS is more complex and involves multiple CAA transporters including system $y^+$, system $B^{0,+}$, system $y^+L$ and system $b^{0,+}$. Overall, this suggests that the intracellular concentration of ADMA is under careful control as multiple pathways are involved in its regulation and this is likely related to its ability to inhibit NO production. In fact, it is possible that ADMA concentration is a critical controller of NO signalling. A major contributor to the transport of both L-arginine and ADMA at both the BBB and BCSFB is the bidirectional transport system, system $y^+$. As system $y^+$ is also a transport system that removes cationic amino acids from cells, it is plausible that this particular transport system, together with the $y^+L$ -system, is involved in the CNS to blood efflux of ADMA that we have previously observed [17]. In this present study we also determined the kinetic constants of [³H]-ADMA delivery into the CNS from the blood. Results suggested that the transporters for ADMA at the BBB and blood-CSF barrier would not be completely saturated by ADMA at the concentrations normally found in the plasma. This would allow these transporters to be utilized by other endogenous CAA that are needed by the brain to function normally. This present study identified two transport systems for ADMA (i.e., system $y^+$ and system $y^+L$) which are involved in ADMA removal from cells and could explain the potential mechanism behind the arginine paradox. We consider that the positive effects of arginine supplementation are due to a reduction in the intracellular ADMA concentration in part as a direct result of transporter interaction (likely system $y^+$). Arginine supplementation in diseases such as Alzheimer's and ischemic stroke is an interesting strategy to improve blood flow and reduce endothelial dysfunction.

## Supporting information

**S1 File. Supplementary information file containing the supporting figures and tables.**
(DOCX)

**S1 Data Set. [³H]-arginine and [¹⁴C]-sucrose in the absence and presence of transporter inhibitors.**
(XLSX)

**S2 Data Set. [³H]-ADMA and [¹⁴C]-sucrose in the absence and presence of transporter inhibitors.**
(XLSX)

**S3 Data Set. [³H]-ADMA and [¹⁴C]sucrose in the absence and presence of unlabelled ADMA at various concentrations.**
(XLSX)

## Acknowledgments

This paper includes data from the PhD thesis of Mehmet Fidanboylu [84].

## Author contributions

**Conceptualization:** Mehmet Fidanboylu, Sarah Ann Thomas.

**Data curation:** Mehmet Fidanboylu, Sarah Ann Thomas.

**Formal analysis:** Mehmet Fidanboylu, Sarah Ann Thomas.

**Funding acquisition:** Sarah Ann Thomas.

**Investigation:** Mehmet Fidanboylu, Sarah Ann Thomas.

**Project administration:** Mehmet Fidanboylu, Sarah Ann Thomas.

**Resources:** Sarah Ann Thomas.

**Supervision:** Sarah Ann Thomas.

**Validation:** Mehmet Fidanboylu, Sarah Ann Thomas.

**Visualization:** Mehmet Fidanboylu, Sarah Ann Thomas.

**Writing – original draft:** Sarah Ann Thomas.

**Writing – review & editing:** Mehmet Fidanboylu, Sarah Ann Thomas.

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
