## [Decision Letter · Decision Letter 0]

15 Apr 2025

Dear Dr. Thomas,

Thank you for submitting your manuscript to PLOS ONE. After careful consideration, we feel that it has merit but does not fully meet PLOS ONE’s publication criteria as it currently stands. Therefore, we invite you to submit a revised version of the manuscript that addresses the points raised during the review process. 

Both reviewers find this manuscript interesting and indicate major revisions.  However, after looking at their comments, I believe that these are addressable in a rather straightforward way (I would label such revisions are minor). 

We look forward to receiving your revised manuscript.

Kind regards,

Alexandra Chittka

Academic Editor

PLOS ONE

Reviewers' comments:

Reviewer's Responses to Questions

**Comments to the Author**

1. Is the manuscript technically sound, and do the data support the conclusions?

Reviewer #1: Yes

Reviewer #2: Yes

2. Has the statistical analysis been performed appropriately and rigorously?

Reviewer #1: Yes

Reviewer #2: I Don't Know

3. Have the authors made all data underlying the findings in their manuscript fully available?

Reviewer #1: Yes

Reviewer #2: Yes

4. Is the manuscript presented in an intelligible fashion and written in standard English?

Reviewer #1: Yes

Reviewer #2: Yes

**Reviewer #1** : The manuscript described transporters for L-arginine and asymmetric dimethylarginine (ADMA) at the blood-brain and blood-CSF barriers. Authors showed that [3H]-arginine predominately uses system-y+, and not system-B0,+, to cross both the BBB and the blood-CSF barrier (choroid plexus). [3H]-ADMA transport at the BBB and blood-CSF barrier can be mediated by system-y+. Thus, these findings will be useful for the treatment of CNS diseases. Therefore, the manuscript is not too excellent to be published. In other words, the manuscript is so excellent that it should be published.

Comments

(1) What was the ratio of the distributed [3H]-L-arginine between the BBB barrier and the CSF barrier?

(2) What was the ratio of the distributed [3H]-ADMA between the BBB barrier and the CSF barrier?

(3) What was the ratio of the distributed [14C]-sucrose between the BBB barrier and the CSF barrier?

(4) Can drug delivery mediated by system-y+ across the BBB be possible using L-arginine or ADMA?

(5) Can drug delivery mediated by system-y+ across the CSF be possible using L-arginine or ADMA?

(6) What features do system y+ has?

(7) What features do system B0,+, has?

(8) What features do system y+L has?

(9) What features do system b0,+ has?

That is all.

**Reviewer #2:**  The authors present a carefully undertaken biochemical study informing carrier systems and their contribution to the transport of arginine and ADMA transported across blood-brain and blood-CSF barriers. The study is valuable and informative and deserves publication. The re-use of previously obtained data is stated and rationalized under the view of ethical frameworks. However the value for clarifying the arginine-paradox and its contribution within disorder context seems a bit of an overstatement. Further comments below.

Abstract:

Why does the involvement of several transport systems for ADMA indicate that it is tightly controlled (in comparison to Arginine). Please extend on this statement (best within discussion).

Line 59 partly ? please complete the information. What do you mean by partly, which are the other mechanisms involved.

Introduction:

Is generally clear and introduces the background, aim and approach of the study. The last sentence is complex and unclear how these contributions to the field will be achieved by the study. Please rewrite.

Methods:

Line 113 same timeframe ? do you mean one batch of experiment? Please clarify.

Please describe the statistical tests used in the paragraph statistics, how where they chosen and if assumptions (normality, equal variance) where tested and how.

Results:

Presentation of results is coherent and complete. Figures are clear and provide information of individual data points and sample size.

Discussion:

Paragraph 1-2 are a repetition/extension of the Introduction. This is redundant and not necessary. Paragraph 3 summary of main findings can also be more concise but would be a good paragraph to start the discussion.

In the abstract and in the conclusion the importance of the presented data for explaining the arginine paradox is stressed. However I find the link weak, please explain the paradox in more detail and how exactly the data contributes to solving it more clearly.

**Do you want your identity to be public for this peer review?** For information about this choice, including consent withdrawal, please see our Privacy Policy

Reviewer #1: No

Reviewer #2: No

---

## [Author Response · Author response to Decision Letter 1]

14 May 2025

We have responded to the reviewers comments in the document labelled 'Response to Reviewers'.

---

## [Editor Report · Decision Letter 1]

19 May 2025

Saturation kinetics and specificity of transporters for L-arginine and asymmetric dimethylarginine (ADMA) at the blood-brain and blood-CSF barriers.

PONE-D-25-07610R1

Dear Dr. Thomas,

We’re pleased to inform you that your manuscript has been judged scientifically suitable for publication and will be formally accepted for publication once it meets all outstanding technical requirements.

Kind regards,

Alexandra Chittka

Academic Editor

PLOS ONE
---

## [Editor Report · Acceptance letter]

PONE-D-25-07610R1

PLOS ONE

Dear Dr. Thomas,

I'm pleased to inform you that your manuscript has been deemed suitable for publication in PLOS ONE. Congratulations! Your manuscript is now being handed over to our production team.

Kind regards,

on behalf of

Dr. Alexandra Chittka

Academic Editor

PLOS ONE